# A Two-Channel Silicon Nitride Multimode Interference Coupler with Low Back Reflection

Jonathan Menahem and Dror Malka *

Faculty of Engineering, Holon Institute of Technology (HIT), Holon 5810201, Israel
* Correspondence: drorm@hit.ac.il

**Abstract:** Optical communication systems based on silicon (Si) multimode interference (MMI) wavelength-division multiplexing (WDM) technology can suffer from back reflection. This undesirable characteristic causes losses and is a key problem that can lead to performance limitations. To overcome this limitation, we proposed a new study on how to divide two wavelengths by understanding the light coupling mechanism of the silicon nitride (SiN) MMI coupler over the C-band window and showed four different options to design a two-channel demultiplexer. The best option for a two-channel SiN MMI coupler with low back reflection losses operating in the C-band spectrum was selected. Based on simulation results, the proposed device can transmit two channels with a spacing of 20 nm between wavelengths in the C-band. Moreover, the device has a low power loss range of 0.895–0.936 dB, large bandwidth of 16.96–18.77 nm, and good crosstalk of 23.5–25.86 dB. Usually, a unique design such as angled MMI is required when using Si MMI technology to reduce the back reflection losses. Due to the use of SiN, which has a low refractive index, we obtained a 40.4 dB back-reflection loss without using this angled MMI design. Therefore, this MMI demultiplexer based on SiN can be used in optical communication systems based on the WDM technique to obtain a high data transfer rate in conjunction with low back-reflection losses.

**Keywords:** SiN; buried waveguide; back reflection; MMI; BPM; PIC; WDM; FDTD

## 1. Introduction

In order to support high-speed light transmission [1] with a wide bandwidth, low losses, and low back reflection losses [2,3], new and powerful waveguide devices are required due to the significant rise in innovations of optical communication systems over the C-window range.

By reducing the distance between peak wavelengths, wavelength-division multiplexing (WDM) technology boosts data transfer rates and allows for the usage of more channels for a single spectral band [4]. A key component is needed for a WDM system to function, which is the (de)multiplexer, to separate/combine the various signals carried within the mode channels and many laser sources for each wavelength.

It is possible to implement optical demultiplexers using a variety of technological approaches. Many advanced multiplexing technologies have been explored, including Mach–Zehnder interferometers [5], multimode interference (MMI) couplers [6–8], and Y-branch devices [9–11].

The basic structure known as a buried waveguide is composed of a narrow high-index zone (strip region) covered by a low-index material which is usually chosen to be silica (SiO2) [12]. Due to the total internal reflection effect, this structure permits light to be tightly contained and steered through it [13]. The main benefit of using a buried waveguide structure is low propagation loss which can improve optical communication system performance [14].

Given the wide optical bandwidth, minimal losses, and small size, MMI couplers are commonly utilized in photonic integrated circuits (PIC) [15–17]. Moreover, due to

its compactness, flexibility in the design of multiport devices, and acceptable fabrication tolerances [18–20], MMI couplers have evolved into a variety of signal-splitting devices for applications in optical communication. The MMI device is a type of waveguide that is designed to support a large number of modes. Access waveguides are positioned at the input and output of that multimode waveguide in order to project light into and retrieve light from it. The self-imaging phenomenon, which is the basis for MMI coupler operation, duplicates the electric field profile that enters the device into numerous images at periodic intervals along the propagation direction of the waveguide [21].

Recent developments in quantum technology and photonic bandgaps may provide a reliable way to transmit entanglement over long distances and potential application with quantum technology [22] and in Lorentzian environments such as in the recent paper [23]. Moreover, phase modulation of coherent states plays a role in quantum communication channels [24] and in the use of probabilistic noiseless linear amplifiers both at the encoding stage [25], where the information is coded on phase shifts, and at the decoding stage [26].

One of the key challenges that may degrade the effectiveness of the transmitter system is back reflection, specifically, the reflection back into the laser source in the opposite direction. Due to the self-imaging phenomenon and the mismatch in the refractive indices of Si and SiO2, reflections may occur in Si-based MMI couplers [27].

According to studies, polycarbonate polymer optical fiber can be used as a multiplexer or demultiplexer for RGB signals with insertion losses (ILs) ranging from 0.6 dB to 1.2 dB [28,29]. A four-channel demultiplexer in the green light spectrum was also implemented using a structure of a multi-slot waveguide based on gallium nitride (GaN) [30]. Moreover, by using a GaN MMI, researchers have been able to divide four [31] and eight [32] channels in the visible and C-band spectrums, respectively. The back-reflection effect, which is crucial for the functioning of the transmitter, was not taken into account in these studies.

The MMI coupler can generally experience different reflections, which can be divided into different types. Internal resonance modes are the first type of reflection, where there are many self-images at once. The second type is a phase mismatch in the MMI input that can cause a reflection back into the access waveguides, which can result in the input being imaged back onto itself.

This problem can be resolved using silicon nitride (SiN), a material with a slightly lower refractive index than Si. Moreover, SiN has a low absorption coefficient and can operate in the C-band spectrum. Additionally, it has been shown for the first time that light coupling analysis of SiN-based MMI coupler waveguides may be implemented as a two-channel demultiplexer device. Furthermore, SiN has two key characteristics when implemented in wavelength de/multiplexers and power splitters in the C-band spectrum [33]: low thermal sensitivity, which means there is only a minor shift in wavelength as the temperature changes by a single degree Celsius [34], and low back reflection. Researchers have demonstrated that the MMI coupler based on SiN has a promising potential [35].

Previous studies have demonstrated that it is possible to reduce ILs by utilizing SiN in a demultiplexer that is based on the Mach–Zehnder interferometer and operates in the O-band [36]; however, there was somewhat significant crosstalk. Back-reflection losses have not been addressed whatsoever.

When fabricating waveguide structures, surface roughness needs to be considered. However, when using SiN material, the surface roughness is smaller than 0.5 nm, which is less than 2.5% of the fabrication tolerance. Moreover, a previous study measured a low optical loss of approximately 1.27 dB/cm for a width of 3 μm [37]. These results make the roughness losses in SiN neglectable.

This study proposes a design for a two-channel wavelength demultiplexer that splits two channels in the C-band light spectrum using an MMI coupler in a SiN buried waveguide structure. It was discovered that the suitable wavelengths were 1540 nm and 1560 nm. Moreover, a further study was made about separating two wavelengths in the C-band

window for four different spacings, which are 10 nm, 20 nm, 30 nm, and 40 nm. This study can be utilized to design multi-channel MMI couplers based on SiN strip waveguides [38].

The device architecture is based on a two-channel MMI coupler alongside a single input waveguide segment and taper, two output tapers, and two s-bends. In order to achieve the self-imaging effect and identify the ideal MMI coupler parameters, the geometrical dimensions of the buried waveguide structure and the MMI couplers were investigated. The beam propagation method (BPM) [39–42] and finite-difference time-domain (FDTD) simulations were run, the results were evaluated, and Python codes were used to assess and process the results.

## 2. Theoretical Aspect and Design of the 1 × 2 Demultiplexer

Figure 1a shows the cross-sectional view at Z = 0 in the XY plane; yellow- and light blue-colored regions indicate the SiN and SiO2 covers, respectively. $H_{strip}$ represents the SiN layer height, and $W_{srtip}$ indicates its width. The refractive indexes of the SiN strip and the SiO2 cover are $n_{strip}$ and $n_{cover}$, respectively, and they are 1.99 and 1.444.

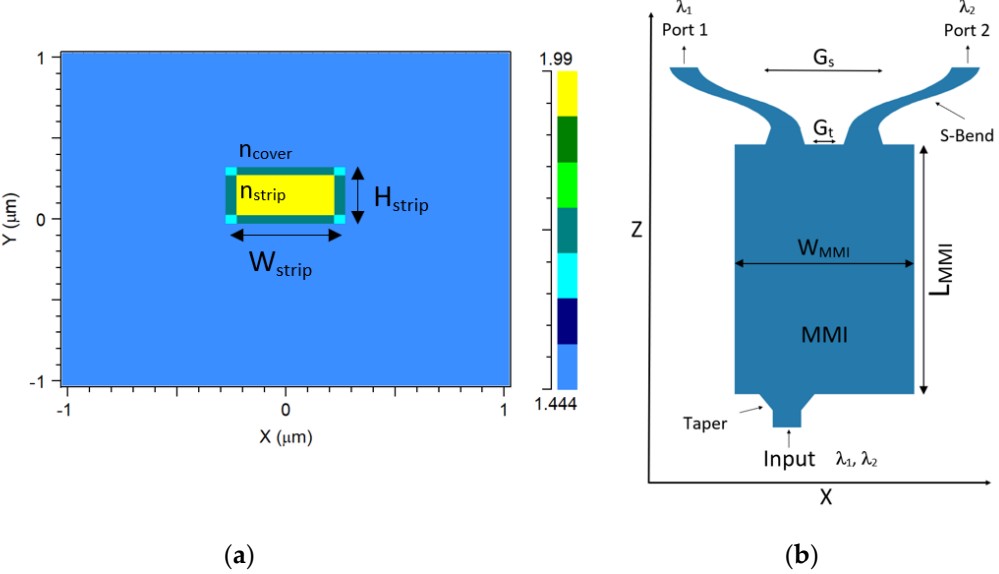

(**a**)　　　　　　　　　　　　　　　(**b**)

**Figure 1.** An illustration of the two-channel MMI demultiplexer. (**a**) The refractive index profile at the XY plane; (**b**) The device profile at the XZ plane.

Figure 1b illustrates the two-channel MMI coupler created for the 1 × 2 demultiplexer device in the XZ plane cross-sectional view at Y = 0. The MMI coupler has a width of $W_{MMI}$ and length of $L_{MMI}$.

The input waveguide segment width and height were set at 500 nm and 320 nm, respectively. For the MMI coupler, the input taper length was specified at 40 μm, and its width ranged from 0.5–0.75 μm. The output taper length was specified to be 35 μm, and its width ranged from 0.9–0.5 μm. $G_t$ and $G_s$ are the gaps between the output tapers and the S-bends and were set to 0.78 μm and 11.5 μm, respectively. The lengths of the S-bends were set to 80 μm.

The two wavelengths that the MMI coupler was designed to divide are 1540 nm (λ1) and 1560 nm (λ2). Figure 1b illustrates how the pair (λ1/λ2) propagates into an MMI coupler before splitting into two channels. The length of the MMI coupler ($L_{MMI}$) is 2067 μm and the width of the MMI ($W_{MMI}$) is 3.5 μm.

The width of the S-bends was set at 0.5 μm to match the width of the output tapers, with a length of 80 μm. The distance between the MMI S-bend outputs ($G_s$) was set at 11.5 μm. The total length of the device is 2.272 mm.

The self-imaging effect states that every wavelength reaching the device's multimode area periodically generates a direct or mirrored picture of itself. The beat length ($L_\pi$) is the distance from the entry to the location of the first image and is provided by [21]:

$$L_\pi \approx \frac{4n_{eff}W_{eff}^2}{3\lambda_n} \; ; \; n = 1, 2 \tag{1}$$

where the operational wavelengths for n = 1, 2 are given by $\lambda_n$.

The SiN effective refractive index for the fundamental electrical mode is known as $n_{eff}$. BPM mode solver was used to determine this parameter. Each fundamental mode has its own "effective" width, also known as $W_{eff}$, which takes into account the horizontal penetration depth due to the Goos–Hanchen shift. The Goos–Hanchen effect is a lateral displacement of a finite cross-sectional wave that is totally internally reflected at the interface between two different refractive index materials. The lateral shift results from the propagation of an evanescent wave parallel to the interface. The $W_{eff}$ size for the transverse electric (TE) mode is provided by [21]:

$$W_{eff} = W_{MMI} + \frac{\lambda_n}{\pi \cdot \sqrt{n_{eff}^2 - n_{cover}^2}} \tag{2}$$

where $W_{MMI}$ is the MMI coupler's actual width, as illustrated in Figure 1b.

This requirement must be satisfied for the MMI coupler to divide different wavelengths:

$$L_{MMI} = pL_\pi^{\lambda_1} = (p + q) \, L_\pi^{\lambda_2} \tag{3}$$

where p is an integer and q is an odd number.

By canceling the third mode from inside the MMI, the propagation distance can be reduced by a factor of three. To achieve this, the MMI input taper needs to be moved out from its center by an offset of $(1/6) W_{eff}$ [21]. Additionally, by employing BPM simulations, these conditions can be optimized to obtain improved performances.

IL can be calculated by using the formula:

$$IL(dB) = -10 \log\left(\frac{P_{out}}{P_{in}}\right) \tag{4}$$

where $P_{in}$ is the power calculated at the input and $P_{out}$ is the power calculated at the output.

Crosstalk losses can be calculated by using the formula:

$$CT_n = 10 \, \log\left(\frac{P_m}{P_n}\right) \tag{5}$$

where $P_n$ represents the power in the desired port and $P_m$ indicates the port power that interferes with the other ports.

The S-bend area size dimensions were carefully selected to reduce bend loss. Zamhari and Ehsan stated that 5 μm is the ideal S-bend offset for Si [43]. As a result, in our situation, the offset is approximately 5 μm, and the following equation can be used to determine the radius of the S-bend:

$$R = \frac{1}{O}\left(\frac{L^2 + O^2}{4}\right) \tag{6}$$

where L is the length of the S-bend and O is its offset.

The MMI coupler was solved using BPM. BPM is generally a very efficient method and has the characteristic that its computational complexity can be optimal in most cases; that is to say, the computational effort is directly proportional to the number of grid points used in the numerical simulation. A commonly used boundary condition is the transparent boundary condition which was used in the BPM simulations for solving the MMI coupler. The back reflection was solved using the FDTD method. The FDTD method is a rigorous

solution to Maxwell's equations and does not have any approximations or theoretical restrictions. This method is widely used as a propagation solution technique in complex integrated optics, especially in situations where solutions obtained via other methods such as the BPM cannot cope with the structure geometry or are not able to calculate the light in the reverse direction. Since FDTD is a direct solution of Maxwell's curl equations, it therefore includes many more effects than other approximate methods. The boundary condition used in the FDTD simulations was the perfectly matched layer, in which both electric and magnetic conductivities are introduced in such a way that the wave impedance remains constant, absorbing the energy without inducing reflections.

## 3. Results

Using BPM and FDTD in Rsoft photonic suite, the MMI coupler and the buried structure were simulated, and the results were processed using Python code to determine the best values.

The TE fundamental mode profile inside the SiN strip for an operating wavelength of 1540 nm is shown in Figure 2a at the XY plane, and its horizontal view, where Y = 0.15 μm, can be seen in Figure 2b. The red hue indicates high confinement, and an adiabatic taper can increase the mode 2.36 × 2.22 μm spot size to improve coupling between the demultiplexer and the laser source. For 1560 nm wavelength, a similar mode profile was obtained. Figure 2c shows the TM fundamental mode solution inside the SiN strip at the XY plane, and its vertical view, where X = 0 μm, is shown in Figure 2d. As can be observed, the TM mode light in the center mode is not entirely confined at only 82%, which can increase the power light losses. Thus, it is better to use the TE mode in our design.

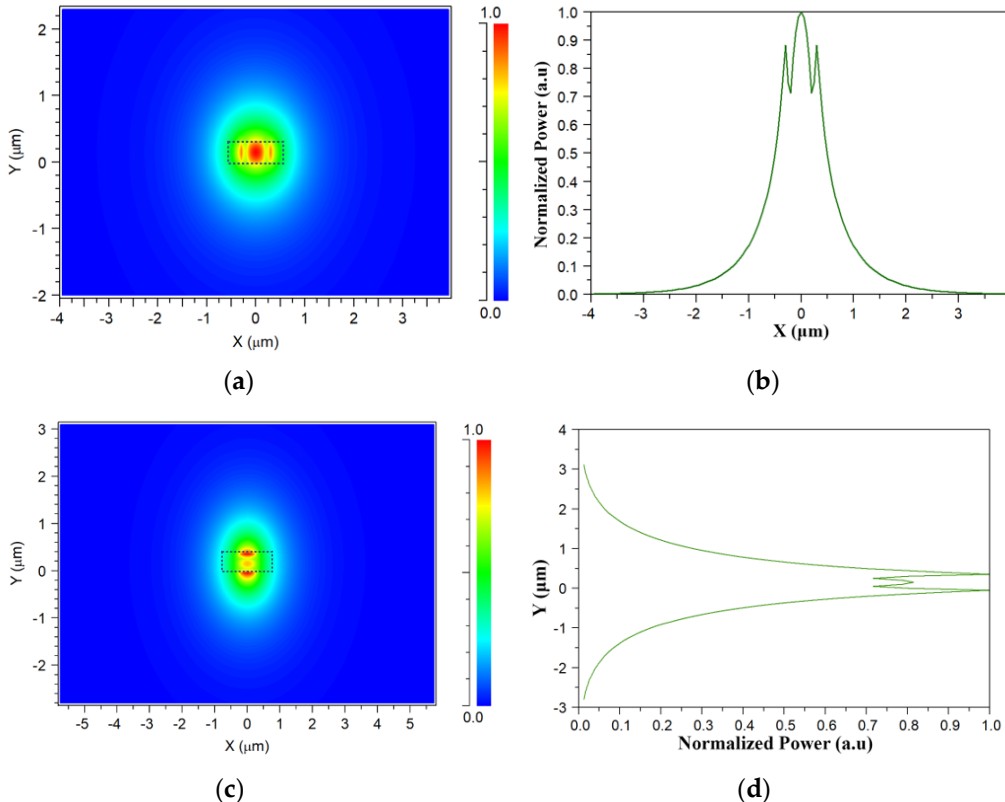

**Figure 2.** (**a**) The 1540 nm TE fundamental mode (SiN area in dotted rectangle) and (**b**) its horizontal view at Y = 0.15 μm; (**c**) The 1540 nm TM fundamental mode (SiN area in dotted rectangle) and (**d**) its vertical view at X = 0 μm.

In order to simulate the device, we used the mode solutions of both wavelengths as the initial launch conditions in the input waveguide of the demultiplexer.

The fundamental mode for each operational wavelength was solved to determine the effective refractive indexes ($n_{eff}$) values, which were 1.4823 and 1.4793 for 1540 nm and 1560 nm, respectively.

Figure 3 illustrates the optimizations for the 320 nm SiN height, which was chosen as the ideal value, and its tolerance range of $\pm 20$ nm ($\pm 6.25\%$) around the ideal value achieved normalized power of 75% to 82% (input power is considered to be 100%). Figure 3 illustrates the maximum power obtained for a height value of 310 nm. However, the maximum fabrication error that can currently be handled is not satisfied by this value from a fabrication standpoint. This restriction is a result of the fabrication process geometrical dimension inaccuracy, which is typically $\pm 20$ nm away from the ideal value. As a result, with a SiN layer thickness error of $\pm 20$ nm, the suggested device can function efficiently.

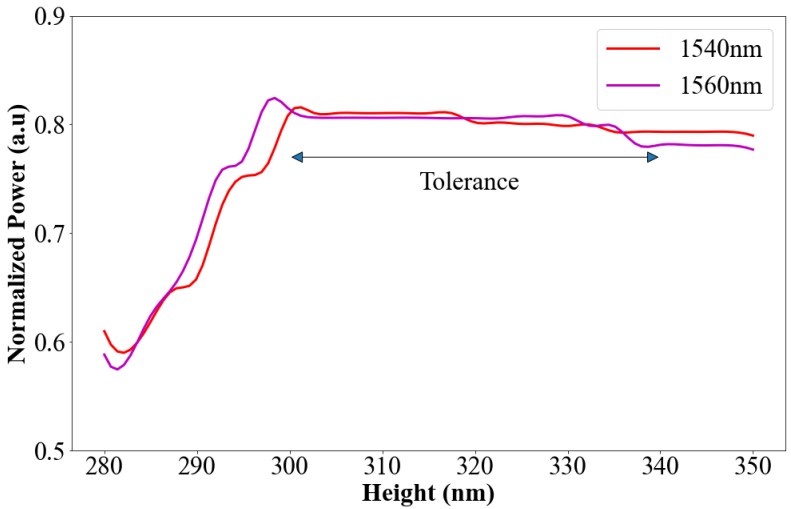

**Figure 3.** Normalized power as a function of SiN strip height.

Figure 4 illustrates the tolerance range of the MMI coupler width, which is $\pm 20$ nm around the ideal value of 3.5 µm ($\pm 0.6\%$), in order to achieve a normalized power (input power is considered to be 100%) of 77% to 81%. This restriction was established because a fabrication process with great accuracy typically deviates roughly $\pm 20$ nm from the ideal value in terms of geometrical dimensions. These fabrication results can be used in the prototype test structure.

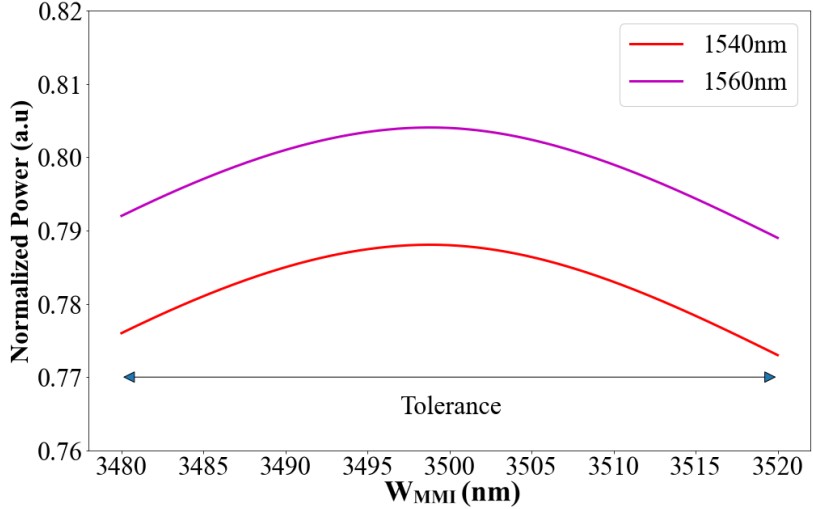

**Figure 4.** Normalized power as a function of MMI width for both wavelengths.

The beat length can be determined by using Equation (1) for each wavelength; the parameter p was found to be 65 using Python code, which takes into account the geometrical parameters and wavelengths and also calculates beat length, efficient MMI width, and the theoretical MMI length, combined with Equations (1)–(3). The parameter p was calculated using the beat lengths of the two selected wavelengths. Using Equation (3), we isolated the parameter p by dividing the smaller beat length by the absolute value of the difference between them. At last, the result was rounded to an integer. Using the calculated parameter p with Equation (3), the theoretical MMI length was determined to be 2072 μm.

The optimal MMI coupler length was determined using Equation (3) and BPM simulation. $L_{mmi}$ was chosen to be 2067 μm to achieve optimum output power with low manufacturing error. Figure 5 shows a 5 μm tolerance range from the optimal value for the MMI length for each of the operating wavelengths in order to achieve normalized power (input power is considered to be 100%) of 70–80% (which is ±0.24% of the optimal value). We have great fabrication flexibility thanks to this wide tolerance range. In other words, a large error dimension around ±250 nm of the MMI coupler length can be easily handled without significantly reducing the power.

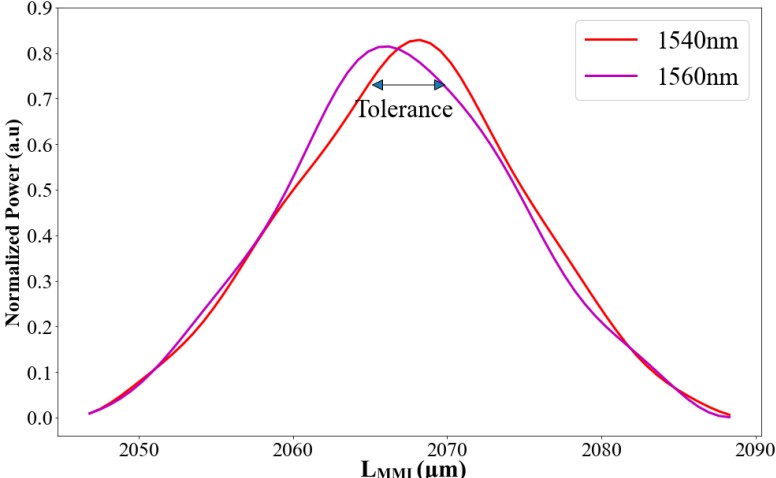

**Figure 5.** Normalized power as a function of MMI length.

These fabrication findings are displayed in Figures 3–5 and can be applied to the prototype test construction to better understand the physical fabrication error of the geometrical parameters ($L_{mmi}$, $W_{mmi}$, and height).

Figure 6a,b illustrates the intensity profile of the two-channel SiN MMI demultiplexer device for the operating wavelengths in the XZ plane. When Z = 0 in the input waveguide segment, the light enters and passes through the input taper before entering the MMI section, and Z = 2157 μm is the point that the two wavelengths split (MMI output). The light propagates through the S-bends separately to the two output ports where Z = 2272 μm. These figures suggest that the SiN MMI coupler has a longer coupling length compared to the Si MMI coupler, resulting in greater footprint size.

Figure 7 illustrates the optical spectrum for the two channels over the C-band range (1535–1565 nm) with a spacing of 20 nm. The data were generated by solving the mode for each wavelength over the C-band with 1 nm intervals and finding the output power for the optimal demultiplexer design. The processing of the data was done by using Python code. As shown in Table 1, the insertion losses, crosstalk, and bandwidth were calculated using Equations (4) and (5) alongside the output power for both ports. The bandwidth was between 16.96 nm and 18.77 nm, the crosstalk was between 23.5 dB and 25.86 dB, and the ILs were between 0.895 dB and 0.936 dB. These findings can be used in a WDM system to double the data bitrate without suffering from crosstalk. For all BPM simulations, the optimal grid sizes for the x, y, and z axes were set to 50 nm and 40 nm, respectively, to achieve high computation accuracy.

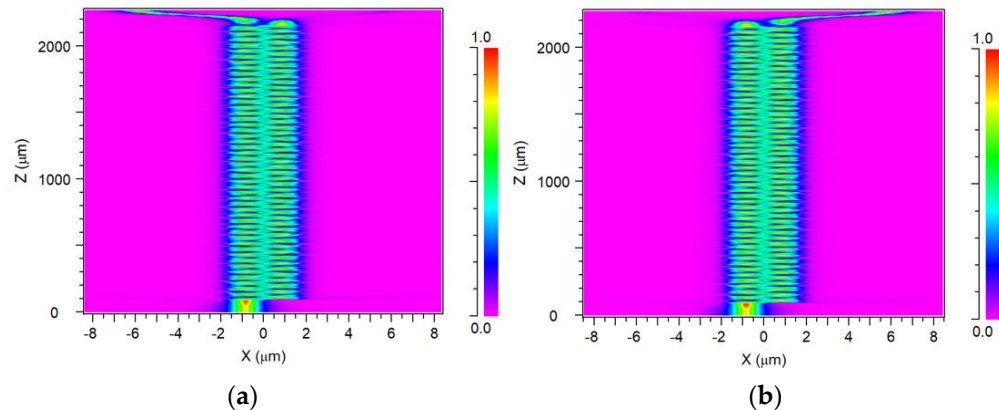

**Figure 6.** Intensity distributions of the two-channel SiN MMI demultiplexer: (**a**) $\lambda_1 = 1540$ nm (port 1); (**b**) $\lambda_2 = 1560$ nm (port 2).

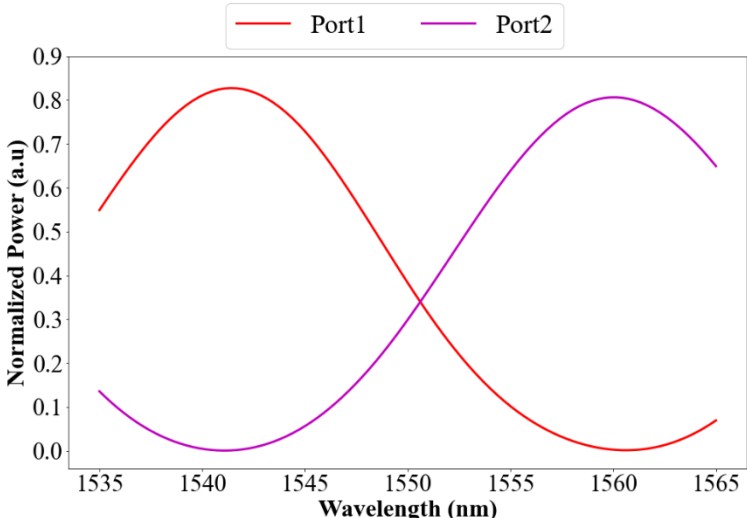

**Figure 7.** Normalized power as a function of wavelength over the C-band range.

**Table 1.** Crosstalk, IL and, Bandwidth values.

| λ (nm) | 1540 | 1560 |
|---|---|---|
| Port | 1 | 2 |
| Crosstalk (dB) | 23.4 | 25.86 |
| IL (dB) | 0.895 | 0.936 |
| Bandwidth (nm) | 16.96 | 18.77 |

To better emphasize the study on SiN MMI coupling parameters and the physical behavior of the coupling length inside the MMI, we investigated the MMI lengths for four different channel spacings over the C-band window. The chosen spacings were 10 nm, 20 nm, 30 nm, and 40 nm. Table 2 shows the wavelength pair values, their corresponding MMI lengths, and ILs for each output channel. It is clear that as the spacing decreases, the device footprint increases, which should be considered when designing a complex, dense WDM system.

**Table 2.** Coupling length behavior for different wavelength spacing channels.

| λ (nm) | 1530–1540 (10 nm Spacing) | 1540–1560 (20 nm Spacing) | 1530–1560 (30 nm Spacing) | 1530–1570 (40 nm Spacing) |
|---|---|---|---|---|
| $L_{MMI}$ (μm) | 4154 | 2067 | 1374 | 1030 |
| IL (dB) | 0.974 | 0.915 | 0.909 | 0.902 |

We chose to use the 20 nm spacing for this device design because it has a good combination of channel separation and is a relatively compact device. This table varies according to the MMI width.

Back reflections are another significant aspect of the MMI coupler. They can be quite harmful to the laser beam source because they can introduce unwanted noise to the system and degrade the signal integrity. In this study, we used SiN material to reduce the MMI coupler self-imaging-related back-reflection power. In order to determine the back-reflection power, a monitor was positioned in the input waveguide to capture all the light that was returning from the MMI coupler, as shown in Figure 8. Table 3 displays the back-reflection losses that were determined by FDTD simulation. As anticipated, the SiN MMI coupler waveguides achieved a decreased back reflection in the C-band spectrum without the need of a special angled MMI. For all FDTD simulations, the ideal x, y, and z axis grid sizes were set to 10 nm to achieve acceptable mesh convergence.

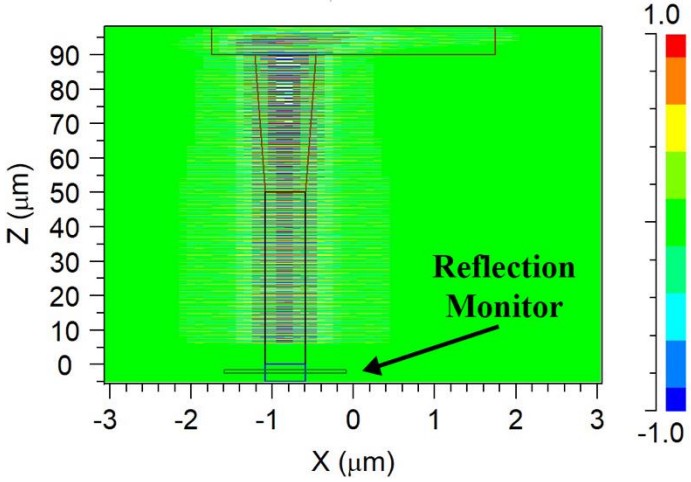

**Figure 8.** Monitor placement illustration in FDTD simulation.

**Table 3.** Back-reflection loss for the operating wavelengths.

| Wavelength (nm) | 1540 | 1560 |
|---|---|---|
| Back Reflection (dB) | −40.8 | −40 |

In order to highlight the benefits of SiN MMI technology over existing demultiplexer designs, comparisons of our design's essential properties to those of previously published studies were made. The SiN MMI demultiplexer design suggested in this work is compared to various types of demultiplexers in Table 4. Insertion losses, crosstalk, bandwidth, band range, back-reflection loss, and device footprint are the main factors that were compared. Table 4 suggests that our solution outperforms existing demultiplexer designs in three categories: lower insertion losses, improved crosstalk, and increased bandwidth. Additionally, the majority of studies do not account for back-reflection loss. Furthermore, our design uses SiN waveguide technology, which is anticipated to have a lower back-reflection loss and can be used with lasers susceptible to the back-reflection phenomenon.

**Table 4.** A comparison of the main factors between various types of demultiplexer designs.

| Demultiplexer Type | Material | Number of Channels | IL (dB) | Crosstalk (dB) | Bandwidth (nm) | Spectral Band | Back Reflection (dB) | Device Footprint (μm²) |
|---|---|---|---|---|---|---|---|---|
| Modified-T [44] | Si | 4 | ~2.3 | ~21.1 | ~0.45 | C | N/A | 536 |
| MMI [45] | Si | 8 | ~3.09 | N/A | N/A | C | ~36.49 | 18 × 18,000 |
| MMI slot waveguide [2] | GaN | 4 | ~0.1 | ~22.7 | ~9.15 | Visible | ~36.49 | 3.8 × 700 |
| Multi-slot waveguide [30] | GaN | 4 | ~0.127 | ~24.1 | ~9.1 | Visible | ~36.5 | 3.2 × 104 |
| MMI buried waveguide [in this work] | SiN | 2 | ~0.915 | ~24.68 | ~17.86 | C | ~40.4 | 12.5 × 2272 |

## 4. Conclusions

The design of a brand new and innovative two-channel demultiplexer in the C-band is presented in this study. It is based on MMI technology and a SiN buried waveguide structure.

Results found the most effective parameters to separate the two wavelengths, which are 1540 nm and 1560 nm.

The device has a 2.272 mm total propagation length, excellent crosstalk of 23.5–25.86 dB, losses of 0.895–0.936 dB, and a bandwidth of 16.96–18.77 nm. These findings indicate that a device of this kind would be advantageous in long-range C-band optical communication networks using the WDM technology to improve the data throughput.

Furthermore, it was demonstrated that using SiN as the core material results in the suggested device has minor back-reflection losses varying between 40–40.8 dB without requiring a unique angled MMI structure.

The findings indicate a promising future for implementing such a device in WDM optical communications systems to improve the data transfer rate.

The suggested device design can be produced using the current fab techniques due to the good tolerance range (5 μm away from optimal value) for the MMI waveguide coupler length.

Because of the straightforward design of the device, adding MMI couplers in series and modifying their geometrical parameters makes it simpler to increase the number of channels.

Although this device was described as a 1 × 2 demultiplexer, it can also be used as a 2 × 1 multiplexer by reversing the direction of the light.

**Author Contributions:** D.M. envisioned the project. D.M. provided guidance. J.M. designed the device. J.M. performed simulations with the support of D.M. and J.M. wrote the paper and the reply letter, J.M. made the figures, and all authors reviewed the manuscript. All authors have read and agreed to the published version of the manuscript.

**Funding:** This research received no external funding.

**Conflicts of Interest:** The authors declare no conflict of interest.

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
