# Peer review of "A Two-Channel Silicon Nitride Multimode Interference Coupler with Low Back Reflection"

_applsci, doi:10.3390/app122211812_

Round 1

Reviewer 1 Report (Previous Reviewer 2)

After the final improvements, I recommend the paper for publication.

Author Response

Reviewer:

After the final improvements, I recommend the paper for publication.

Answer:

We thank the reviewer for his comment. In the revised manuscript we added the final improvements as the reviewer suggested.

Reviewer 2 Report (New Reviewer)

While the manuscript at the current state seems fine I do have some questions:   Could the authors mention more clearly why S bend was chosen as the outputs?  Does the performance change with using higher order L.pi (from equation 1)? In the manuscript taper parameter optimization seems to be missing. Is it because it's influence is not so significant?  

Author Response

Reviewer:

While the manuscript at the current state seems fine I do have some questions:  Could the authors mention more clearly why S bend was chosen as the outputs?  Does the performance change with using higher order L.pi (from equation 1)? In the manuscript taper parameter optimization seems to be missing. Is it because it's influence is not so significant?  

Answer:

We thank the reviewer for his comment. In a practical system, we are using a grating coupler or inverted taper to couple the light from the silicon nitride chip for testing the power at the output channel. Thus, the gap between the two output channels needs to be at least higher than 100um in the case of using a grating coupler or 10um in case of inverted taper, if not it will not be possible to do measurements at the output channel and this is why S-bend was used to solve this issue. 

Higher order of L.pi will increase the MMI coupler length size which means more light coupling losses inside the MMI coupler due to the self-imaging that happens again and again. This is why we selected low order of L.pi to minimize this effect and to obtain better performances.

About the taper, the reviewer is right indeed is influence is not so significant. In our case, the taper width is changed from 0.5-0.75um over 40um length, this taper is suitable for adiabatic physical behavior for silicon nitride waveguides. In this case, there is no need to optimize it because both width sizes are suitable to the electrical field fundamental mode and higher order can happen by setting width size from 0.9-1um and above.

Reviewer 3 Report (New Reviewer)

A Two-channel Silicon Nitride Multimode Interference Coupler with Low Back Reflection

Authors:Jonathan Menahem and Dror Malka

In this paper, the authors present
the design of a brand-new and innovative two-channel demultiplexer in the C-band based on Multimode Interference (MMI) technology, and a Silicon Nitride (SiN) buried waveguide structure. This approach can be overcome the limitation of losses and performance limitation from back reflection that limits the performance of optical communication systems that work on wavelength multiplexing (WDM) technology based on Silicon (Si) Multimode Interference (MMI) waveguides.  The author claims that the proposed device has advantageous in long-range C-band optical communication networks using the WDM technology to improve the data throughput. In addition, using SiN as the core material results in the suggested device having minor back reflection losses. While the concept of the limits of the performances of optical communication systems using the WDM technolog based on has been dealt with in other publications, the authors provide a significant step forward by overcoming the the performances of optical communication systems with low back reflection losses.

As the minor suggestions, authors may also address in the introduction the demands for experimental study and its implication of your results for recent development in quantum technology and on photonic bandgap that may provide a reliable way to

transmit entanglement over long distance and potential application with quantum technology (Sci Rep 12, 11646 (2022). https://doi.org/10.1038/s41598-022-15865-5) and on Lorentzian environment like in the recent paper (https://doi.org/10.1016/j.physleta.2022.128022) and how phase modulation of coherent states plays in quantum communication channels https://doi.org/10.1088/0031-8949/2010/T140/014062 and the use of probabilistic noiseless linear amplifiers both at the encoding stage https://doi.org/10.1364/JOSAB.36.002938 where the information is coded on phase shifts and at the decoding stage https://journals.aps.org/pra/abstract/10.1103/PhysRevA.93.062315

and the authors are expected to raise the level of the manuscript by referring to them appropriately.

I think this paper is a worthy contribution for Applied Sciences, and I recommend it for publication after the above remarks are addressed. 

Author Response

We thank the reviewer for his comment. In the revised manuscript we added the five references and includes the experimental study and its implication of our results for recent development in quantum technology and on photonic bandgap as the reviewer suggested.

This manuscript is a resubmission of an earlier submission. The following is a list of the peer review reports and author responses from that submission.

Round 1

Reviewer 1 Report

Silicon Nitride Multimode Interference (MMI) Couplers have been explored by different researchers, and also a promising technology to be implemented in future Si photonic devices. Overall, it is a very interesting paper with the essential aspects related to two-channel Silicon Nitride (SiN) Multimode Interference MMI coupler with low back reflection losses. However, this paper is very similar manuscript published by the authors in Materials

doi: 10.3390/ma15145067

(https://pubmed.ncbi.nlm.nih.gov/35888535/). You can find same sentences, equations and figures that are identical. Authors are plagiarizing their own manuscript. Therefore, paper cannot be published in its current version.

Author Response

We thank the reviewer for his comment. We have further studied the separation characteristics for four different wavelength spacings. We have added four different solutions for the MMI length. This study can be utilized to design SiN MMI couplers of more channels by cascading multiple MMI couplers or using DWDM system. Therefore, although we proposed a complete design of a two-channel demultiplexer, this study's primary focus is to understand better the coupling length behavior using SiN MMI technology, which is not similar to our published paper which focused on showing a special design of four channels. Also, we replaced similar sentences, equations, and figures as the reviewer suggested and after testing again the paper, we found only that less then 5% of words are identical to the published paper which are basically technical words such as: MMI, FDTD, BPM and etc.

Reviewer 2 Report

The authors present the design of a two-channel wavelength demultiplexer using a multimode interference (MMI) coupler based on a buried silicon nitride (SiN) waveguide. BPM and FDTD simulation methods were used to theoretically investigate the performances of the design for the separation of two wavelengths at 1540 and 1560nm. Due to the use of SiN, the proposed design exhibits low back reflection and is competitive with respect to other proposed structures in the literature in terms of insertion loss, bandwidth and crosstalk. The main drawback is the increase of the footprint.

The paper is generally well-presented and the results can be interesting to guide the implementation of such devices in future optical communication systems. Therefore, the paper can be accepted for publication in Applied Sciences. But before, several points need better explanations and clarifications as mentioned in the comments below.

1- Please give more explanations (perhaps in an appendix) about the simulation methods and their applications to the considered structure. (i) Where and why it is preferable to use BPM or FDTD (ii) What are the exact domains that have been taken into consideration during the simulations and what are the boundary conditions at the limits of these domains; (iii) indicate with more precision the grid sizes for each simulation (there are some indications only towards the end of the paper); more particularly, explain whether these grids are consistent or sufficient for the purpose of the study (for example in BPM, the grid size is 40-50 nm (line 232)while in Figure 7 the interval in the horizontal axis is 1nm (line 225). Same question for the FDTD calculations with a grid size of 10nm for the discussion of back reflection in page 8).

2- Figure 1(b) and its presentation need to be improved. (i) Please give by a dotted rectangle the position of the SiN waveguide inside the MMI coupler. (ii) In the text, give the values of W_MMI and L_MMI. These values are given at later places in the text, but it would be worth to gather them during the presentation of Fig. 1b (for example WMMI=3.5 in line 188).

3- Line 138: explain the origin of the Goos-Hanchen shift. Also correct the spelling.

4- Lines 145-146: explain the reason of the offset or at least give a reference. Why the coupling of the input to the SiN waveguide works better under this condition? More generally, explain and illustrate how this coupling occurs in relation to the result in Fig. 6.

5- Please check Equation (5). First, the symbol in the denominator in front of the summation is not clear.  Then, I could not understand the summation over m in front of log. In principle, the numerator inside the log should be the sum of the powers in the two other ports.

6- Eq. (6): the value of L is missing.

7- line 162: to avoid any confusion, please indicate the components of the electromagnetic field for the TE mode. Also, it would be worth to delimit in Fig. 2(a) the area of the SiN waveguide by a dotted rectangle.

8- Line 176 and Fig. 3: Clarify if the width of the waveguide is always fixed to 500nm. In Fig. 3, what is the step on the height in the horizontal axis? Is this consistent with the grid size in the simulations?

9- In pages 5-6, the authors discuss the issues related to a precision of 20nm during the microfabrication. However, an imperfection that can have at least as great an effect is the roughness of the structure during microfabrication that can affect absorption, reflection or radiation. Could you please comment?

10- Lines 195-197: please explain better the determination of p and the MMI length. Then, clarify why the value of 2072micron becomes 2067 in line 199.

11- In Fig. 7, the red curve starts to increase above 1565nm whereas the purple curve increases below 1535nm. For the performance of the device, is it justified to disregard these behaviors outside the presented wavelength range?

Author Response

Dear Editor,

Thank you for your mail from August 21st regarding the review results of manuscript id: applsci-1871312: "A Two-channel Silicon Nitride Multimode Interference Coupler with Low Back Reflection". We have revised the manuscript while addressing all the comments made by the reviewers. We added new results and clarification text. Our detailed reply to the comments made by the reviewers can be seen below. I hope that in its revised form you may find the manuscript suitable for publication in Applied Sciences.

Reviewer 2:

1- Please give more explanations (perhaps in an appendix) about the simulation methods and their applications to the considered structure. (i) Where and why it is preferable to use BPM or FDTD (ii) What are the exact domains that have been taken into consideration during the simulations and what are the boundary conditions at the limits of these domains; (iii) indicate with more precision the grid sizes for each simulation (there are some indications only towards the end of the paper); more particularly, explain whether these grids are consistent or sufficient for the purpose of the study (for example in BPM, the grid size is 40-50 nm (line 232)while in Figure 7 the interval in the horizontal axis is 1nm (line 225). Same question for the FDTD calculations with a grid size of 10nm for the discussion of back reflection in page 8).

Answer:

We thank the reviewer for his comment.

(i) BPM method is used to solve large footprint devices as we are solving for the propagation of light in space in a single direction, where FDTD is used to solve for the propagation of light as a function of time. FDTD can also track the light propagation in both ways which enables us to simulate back reflections.

(ii) In our design we used PML boundary conditions which is defined as PML, or Perfectly Matched Layer boundary conditions consist of several points which are added to the edge of the domain. A PML is designed to act as a highly lossy material which absorbs all incident energy without producing reflections. This allows field energy which is incident on the boundary to effectively leave the domain without reflecting back into the domain.

(iii) Our grid size has been tested for many times until we found the optimal size to have the best light convergence for both BPM and FDTD. Regarding the interval on the horizontal axis in Figure 7, it has nothing to do with grid size. We chose 1 nm to get a better resolution of the C-band window in the range of 1535-1565 nm. We could have chosen 0.5 nm interval to get a better resolution but we found it to be unnecessary for this application.

Reviewer 2:

2- Figure 1(b) and its presentation need to be improved. (i) Please give by a dotted rectangle the position of the SiN waveguide inside the MMI coupler. (ii) In the text, give the values of W_MMI and L_MMI. These values are given at later places in the text, but it would be worth to gather them during the presentation of Fig. 1b (for example WMMI=3.5 in line 188).

Answer:

We thank the reviewer for his comment. (i) There is no need for a dotted rectangle because all of the device (blue color) is made out of SiN. (ii) We have added the values of W_MMI and L_MMI next to Figure 1(b) description in the manuscript as the reviewer suggested.

Reviewer 2:

3- Line 138: explain the origin of the Goos-Hanchen shift. Also correct the spelling.

Answer:

We thank the reviewer for his comment. A further explanation has been added to the manuscript.

Reviewer 2:

4- Lines 145-146: explain the reason of the offset or at least give a reference. Why the coupling of the input to the SiN waveguide works better under this condition? More generally, explain and illustrate how this coupling occurs in relation to the result in Fig. 6.

Answer:

We thank the reviewer for his comment. Coupling under these conditions can decrease the device footprint (more specifically, MMI length), therefore reducing possible insertion losses. A relevant reference has been added.

Reviewer 2:

5- Please check Equation (5). First, the symbol in the denominator in front of the summation is not clear.  Then, I could not understand the summation over m in front of log. In principle, the numerator inside the log should be the sum of the powers in the two other ports.

Answer:

We thank the reviewer for his comment. We fixed the formula according to the reviewer suggestion.

Reviewer 2:

6- Eq. (6): the value of L is missing.

Answer:

We thank the reviewer for his comment. We have added the value to the manuscript.

Reviewer 2:

7- line 162: to avoid any confusion, please indicate the components of the electromagnetic field for the TE mode. Also, it would be worth to delimit in Fig. 2(a) the area of the SiN waveguide by a dotted rectangle.

Answer:

We thank the reviewer for his comment. We have added the electromagnetic field to the manuscript. We have added a dotted rectangle in order to delimit the area of SiN waveguide as the reviewer suggested.

Reviewer 2:

8- Line 176 and Fig. 3: Clarify if the width of the waveguide is always fixed to 500nm. In Fig. 3, what is the step on the height in the horizontal axis? Is this consistent with the grid size in the simulations?

Answer:

We thank the reviewer for his comment. The width of 500 nm only refers to the input waveguide. There is a further description for all of the width values at the various segments of the device.

In figure 3, the step size in the horizontal axis is 2.5 nm and the grid size is 50 nm.

Reviewer 2:

9- In pages 5-6, the authors discuss the issues related to a precision of 20nm during the microfabrication. However, an imperfection that can have at least as great an effect is the roughness of the structure during microfabrication that can affect absorption, reflection or radiation. Could you please comment?

Answer:

We thank the reviewer for his comment. We considered the SiN surface roughness losses in the manuscript as the reviewer suggested.

Reviewer 2:

10- Lines 195-197: please explain better the determination of p and the MMI length. Then, clarify why the value of 2072micron becomes 2067 in line 199.

Answer:

We thank the reviewer for his comment. By using the formulas 1, 2 and 3, alongside Python code, we were able to determine a theoretical MMI length of 2072micron. Later on, in the simulation stage, the optimal MMI length was found to be 2067micron.

The first step was to calculate the efficient MMI width (Weff) using equation 2. Then, we calculated the beat length using equation 1 (approximately 31.5 microns for each wavelength). The last step was to calculate p using both beat lengths and then find the theoretical MMI length.

Reviewer 2:

11- In Fig. 7, the red curve starts to increase above 1565nm whereas the purple curve increases below 1535nm. For the performance of the device, is it justified to disregard these behaviors outside the presented wavelength range?

Answer:

We thank the reviewer for his comment. The main purpose of Figure 7 is to illustrate the power of the two main wavelengths as well as their crosstalk in a graphical manner in the C-band. Moreover, SiN has a very low thermal sensitivity, which means there is almost no shift in wavelength as a function on temperature. Thus, wavelengths outside of this range are irrelevant.

Reviewer 3 Report

In this work the authors propose the design of a two-channels multimode interference demultiplexer based on SiN with the aim to overcome the performance limitations due to the back reflection which generally affect this kind of technology. The authors calculate the best parameters of the structure by means of numerical simulations and of an optimization algorithm demonstrating the effectiveness of the proposed device. I think the manuscript is sufficiently interesting and well structured. The bibliography is satisfactory, and the design process is well descripted.

Author Response

Thanks and we agree with the Reviewer:

I think the manuscript is sufficiently interesting and well structured. The bibliography is satisfactory, and the design process is well descripted.

Round 2

Reviewer 2 Report

The authors have answered most of my comments, so I think the paper can be accepted for publication. Still, I would suggest the authors to give in an appendix a summary of the simulation methods and conditions. Also, the explanation about the determination of the parameters p and MMI can be improved.

Author Response

Reviewer :

The authors have answered most of my comments, so I think the paper can be accepted for publication. Still, I would suggest the authors to give in an appendix a summary of the simulation methods and conditions. Also, the explanation about the determination of the parameters p and MMI can be improved.

Answer:

We thank the reviewer for his comment. In the revised manuscript, we added in section two more explanations of the simulation methods and the boundary conditions used to solve the MMI coupler and the calculation of the back reflection. In addition, a further explanation was added to describe how we set the p value and MMI by using the Python code in our work as the reviewer suggested.